# TiO_2_ Decorated Low-Molecular Chitosan a Microsized Adsorbent for a ^68^Ge/^68^Ga Generator System

**DOI:** 10.3390/molecules26113185

**Published:** 2021-05-26

**Authors:** Jun-Young Lee, Pyeong-Seok Choi, Seung-Dae Yang, Jeong-Hoon Park

**Affiliations:** Korea Atomic Energy Research Institute, Jeongeup-Si 56212, Korea; ljy01@kaeri.re.kr (J.-Y.L.); cps@kaeri.re.kr (P.-S.C.); sdyang@kaeri.re.kr (S.-D.Y.)

**Keywords:** ^68^Ge/^68^Ga generator, adsorbent, chitosan-TiO_2_, acid resistance, ^68^Ga-radiopharmaceuticals

## Abstract

We report column material for a ^68^Ge/^68^Ga generator with acid resistance and excellent adsorption and desorption capacity of ^68^Ge and ^68^Ga, respectively. Despite being a core element of the ^68^Ge/^68^Ga generator system, research on this has been insufficient. Therefore, we synthesized a low molecular chitosan-based TiO_2_ (LC-TiO_2_) adsorbent via a physical trapping method as a durable ^68^Ge/^68^Ga generator column material. The adsorption/desorption studies exhibited a higher separation factor of ^68^Ge/^68^Ga in the concentration range of HCl examined (0.01 M to 1.0 M). The prepared LC-TiO_2_ adsorbent showed acid resistance capabilities with >93% of ^68^Ga elution yield and 1.6 × 10^−4^% of ^68^Ge breakthrough. In particular, the labeling efficiency of DOTA and NOTA, by using the generator eluted ^68^Ga, was quite encouraging and confirmed to be 99.65 and 99.69%, respectively. Accordingly, the resulting behavior of LC-TiO_2_ towards ^68^Ge/^68^Ga adsorption/desorption capacity and stability with aqueous HCl exhibited a high potential for ion-exchange solid-phase extraction for the ^68^Ge/^68^Ga generator column material.

## 1. Introduction

Gallium-68 (^68^Ga) is a positron-emitting radionuclide obtained on-site using a ^68^Ge/^68^Ga generator without the need for an accelerator. ^68^Ga has a short half-life (t_1/2_; 67.71 min), decaying 89% through positron emission with a maximum energy of 1899.1 keV and a mean of 836.0 keV [1]. In human positron emission tomography/computed tomography (PET/CT), images reconstructed with PET spatial resolution phantom (^68^Ga) have shown a resolution of 7.0 mm full width at half maximum (FWHM) [2], which is suitable for clinical-based scanning. Therefore, ^68^Ga is a positron-emitting radionuclide that can be used equally with halogen radioisotopes (RIs) such as fluorine-18 and iodine-124. ^68^Ga-based radiopharmaceuticals are increasingly being used in nuclear medicine worldwide for PET/CT tests [3,4]. Over the past few decades, clinical PET imaging research using ^68^Ga has increased tremendously [5]. ^68^Ga-radiopharmaceuticals have been recognized for the diagnosis of bone infection and tumors. For example, ^68^Ga-labeled somatostatin receptor-specific peptides (SSTRs) [6], 1,4,7,10-tetra-azacyclododecane-1,4,7,10-tetraaceticacid-D-Phe^1^-Tyr^3^-octreotide (DOTATOC) in the European Union and DOTA-Tyr^3^-octreotate (DOTATATE) in the United States, are being used successfully for PET imaging [7,8].

^68^Ge/^68^Ga generators are commercially available worldwide, and their use in the medical field is increasing rapidly [9]. Therefore, researching ^68^Ge/^68^Ga generator column materials and studying the capability of adsorption and desorption of ^68^Ge and ^68^Ga radionuclides represent an urgent scientific need of the involved community. Most column adsorbents acting as the core elements of the ^68^Ge/^68^Ga generator use metal oxides [10]. In general, metal oxides have good adsorption capacities for metal ions and provide high surface area with varying surface functional groups that can interact with heavy metal ions [11,12]. As a result, they show remarkable potential for the removal of heavy metal ions. However, the metal oxide adsorbents are poor in acid-resistance [13]. A prefiltration process of eluted Ga-68 is performed to overcome this limitation. Different types of metal oxides have been intensively investigated for ^68^Ge/^68^Ga generator systems, such as SnO_2_, SiO_2_, TiO_2_, ZnO_2_ and Al_2_O_3_ [14,15,16]. However, such adsorbents have low acidic stability and high ^68^Ge breakthrough, which deteriorates over time.

Chitosan is a natural biopolymer obtained by the deacetylation of chitin. It is the main structural component of crustacean exoskeletons [17,18,19]. It is a biocompatible material and is harmless to the human body even when used as an inert support for adsorbents [20]. In particular, chitosan in the presence of amino and hydroxyl functional groups can react with metal ions by electrostatic forces and hydrogen bonds [20,21]. Furthermore, it is possible to increase the capacity of capturing metal ions by exerting a synergistic effect by titanium trapped on the chitosan polymeric matrix [17]. For the removal of heavy metal ions from wastewater, studies have been conducted based on chitosan-adsorbents. Still, as a material for adsorption/separation of RI generators, studies considering elution pressure, acid resistance, radiation resistance, and adsorption and desorption efficiency of RIs represent a new approach [22,23].

In a previous paper [24], we described the adsorption and desorption capacity of ^68^Ge/^68^Ga by synthesizing spherical chitosan-TiO_2_ of 700 μm. Compared to these results, we expect to improve the adsorption and desorption capacity of ^68^Ge/^68^Ga and acid-resistance by adjusting the size of the chitosan-TiO_2_ adsorbent using low molecular chitosan.

In this study, low molecular chitosan and titanium dioxide (P25) were used to develop ^68^Ge/^68^Ga generator column material to ensure the high adsorption/desorption capacity of ^68^Ge (parent nuclide)/^68^Ga (daughter nuclide), respectively, while maintaining its stability in hydrochloric acid, which is generally used as a ^68^Ga eluent (Scheme 1).

## 2. Results

### 2.1. Preparation and Characterization of LC-TiO_2_

Low molecular chitosan and titanium dioxide were used as an inert support and stationary phase to synthesize a microsized LC-TiO_2_ adsorbent with excellent adsorption/desorption capability for ^68^Ge/^68^Ga, while securing stability towards corrosive HCl. The reaction of viscous acetic acid-based low molecular chitosan with titanium dioxide (P25) precursor formed a chitosan-titanium composite-based polymeric matrix. (Figure 1).

The synthesized LC-TiO_2_ adsorbent was characterized for particle size, surface morphology and elementary composition with a scanning electron microscope-energy dispersive X-ray spectrometer (SEM-EDX) and elemental mapping system. SEM images of LC-TiO_2_ adsorbent are shown in Figure 2. In Figure 2a, the average particle size of the synthesized LC-TiO_2_ is around 250 ± 30 μm. At higher magnification (Figure 2b,c), the TiO_2_ trapped chitosan polymer was confirmed, which was physically bonded in a polymeric structure. Besides, elemental mapping showed that titanium nanoparticles were evenly distributed in the LC-TiO_2_.

Figure 3a shows the X-ray diffraction (XRD) pattern of LC-TiO_2_ adsorbent prepared by a mechanical stirring method. The XRD peaks can be perfectly indexed to the anatase and rutile structures of TiO_2_ (COD No. 9015929 and ICDD No. 01-088-1172, respectively). The average crystallite size (Å) of LC-TiO_2_ adsorbent can be estimated according to the diffraction and reflection using the Williamson-Hall method. The crystallinity of anatase and rutile phases was confirmed, which has an individual crystal size of 167 and 653 Å. The crystal phase and size were similar to P25, reflecting the physical sorption to the chitosan polymeric template without modification of P25 during synthesis.

The nitrogen adsorption-desorption (BET) isotherm (Figure 3b) of LC-TiO_2_ adsorbent was measured. The pore size distribution plot was measured using the Barrett–Joyner–Halenda (BJH) method from the desorption branch of the isotherm. The physical properties were measured with a high surface area of 36.05 m^2^·g^−1^, pore-volume and pore size of 0.13 cm^3^·g^−1^ and 14.52 nm, respectively.

### 2.2. Acid Resistance

Metallic impurities (Table 1) were measured after immersing LC-TiO_2_ in 0.05 M HCl eluent. The concentration of titanium ion was confirmed to be 4 ppb. The level of titanium was too low, proving the mechanical stability of the synthesized composite. Considering the material stability and the half-life of ^68^Ge (parent radionuclide) of ~271 days, the column material can be used as a ^68^Ge/^68^Ga generator for >1 year. Besides, the concentration of other metallic impurities such as Sc, V, Cr, Mn, Zn, Ru, Rh, Pd, Ag, and Cd were determined in immersed LC-TiO_2_ with 0.05 M HCl and found to be at extremely low levels and always under 15 ng·mL^−1^ (ppb) as shown in Table 1.

### 2.3. Distribution Coefficient

The distribution coefficient of ^68^Ge(IV)/^68^Ga(III) ions on the LC-TiO_2_ adsorbent at different molarity of HCl are displayed in Figure 4. The *K_d_* value of parent radionuclide ^68^Ge(IV) was maximum at pH 2 while the ^68^Ga(III) was less than 1 mL·g^−1^ at 0.05 M HCl. In theory, at concentrations lower than 0.01 M HCl, the *K_d_* value should increase by hydrolysis of the Ga ion [25]. The Ga(III) ion changes from Ga^3+^→Ga(OH)^2+^→Ga(OH)_2_^+^→Ga(OH)_3_ to Ga(OH)_4_^−^, which is expected for the reduced desorption capacity of the Ga (III) ion. In addition, ^68^Ge(IV) was evaluated for having high adsorption capacity (*K_d_*; 3512 ± 0.630 mL·g^−1^). The ^68^Ge/^68^Ga separation factor (SF) using metal oxide is about ~10^2^. In particular, the SF for titanium dioxide is ~230. In contrast, the adsorbent developed in this study showed that was more than 1000 time higher.

### 2.4. Column Study

Evaluation of the adsorption capacity of ^68^Ge depending on time is required for preparing the ^68^Ge/^68^Ga generator column material. The adsorption yield of ^68^Ge reached the maximum capability within 3 h (Figure 5). The elution profile (Figure 6) of ^68^Ga depended on the HCl concentration. >93% of the ^68^Ga radioactivity was eluted with 0.05M HCl in the 1 mL volume due to the high separation factor of ^68^Ge/^68^Ga. The breakthrough of ^68^Ge was confirmed to be 1.6 × 10^−4^%. This result guarantees the quality of ^68^Ga-radiopharmaceuticals.

### 2.5. Radiolabeling and In Vivo Evaluation

After the reaction, radiochemical yields% (RCY%) of ^68^Ga-DOTA and ^68^Ga-NOTA were 99.65% and 99.69%, respectively. Experimental data are supplied for each chelator in Figure 7a–c.

In vivo small-animal PET studies using ^68^Ga-DOTA and ^68^Ga-NOTA (Figure 7b) show the possibility of introducing ^68^Ga eluted from the column into radiopharmaceuticals. Radiolabeled DOTA and NOTA chelators are potential candidates for the diagnosis of various diseases and cancer. Radiolabelled ^68^Ga exhibited high accumulation in kidney and low uptake in normal tissues in a mouse model, as shown in Figure 7.

## 3. Discussion

The LC-TiO_2_ adsorbent was synthesized by a physical deposition method. This method is effective in enhancing the trapping of titanium dioxide nanoparticles in the polymeric chitosan. The synthesized LC-TiO_2_ was characterized by SEM, EDX, XRD and BET equipment.

The size and morphology of the adsorbent were confirmed through SEM analysis. LC-TiO_2_ was synthesized to enhance the conditions (elution pressure, stability, adsorption/desorption capacity) necessary for its use as a column material for a ^68^Ge/^68^Ga generator system. Compared to the 700 µm sized chitosan-TiO_2_ adsorbent in our previous study [24], spherical LC-TiO_2_ of 250 µm was synthesized, which has low elution pressure and high reactivity between the eluent and the adsorbent. In the EDX and mapping element analysis results, it was speculated that TiO_2_ could effectively react with ^68^Ge and ^68^Ga metal ions and confirming that Ti is evenly distributed in the chitosan template.

The TiO_2_ used in this study was Degussa P25 of about 21 nm commercial size, with a mixture of about 80% anatase and 20% rutile in XRD measurements. After a physical trap reaction of low molecular chitosan and TiO_2_, it was confirmed that the homozygous polymorphic form of anatase and rutile was maintained. Titanium tetraisopropoxide (TTIP) used as a Ti source other than P25 can be hydrolyzed and undergo a phase transition from an anatase phase to a rutile structure even at low temperatures by H^+^ generated as a by-product. It is known that the ratio of the rutile phase increases [26].

Figure 3b shows the N_2_-sorption graph and pore distribution obtained through BET measurement of LC-TiO_2_. The BET-specific surface area of the LC-TiO_2_ adsorbent prepared by the physical trap method was measured to be 36.05 m^2^·g^−1^. From the N_2_-sorption graph in 3(b), it can be seen that pores of 14.52 nm exist due to the difference in the adsorption and desorption curves occurring at the point where the relative pressure is about 0.6.

In the acid resistance results, the content of metal impurities was measured using ICP-MS (inductively coupled plasma-mass spectrometer) equipment. Stability evaluation of the elution pressure of ^68^Ga and hydrochloric acid as eluent was performed to determine whether it could be introduced as a column material for a ^68^Ge/^68^Ga generator. Since a low-molecular chitosan-Ti based adsorbent is used, stability in an acid solution against Ti metal ions must be secured. In general, Ti^4+^ ion can be derived from Ti metal using acid, during which there is a migration of H^+^ ions from the acid to the metal surface, forming H^+^ and H_2_ [13]. The analysis of metal element content confirmed that background-level elements were included, and further studies are being conducted to secure long-term stability.

The distribution coefficient of ^68^Ge and ^68^Ga on LC-TiO_2_ at different molarities of HCl is displayed in Figure 4. The adsorption capacity of ^68^Ge was maximum at 0.01 M HCl, and the trend continued to decrease. The *K_d_* value of ^68^Ga is shown, which had a slight difference, and a similar level of high desorption ability above a concentration of 0.05 M HCl.

Considering the distribution coefficient results of ^68^Ge and ^68^Ga, a column study was performed using 0.05 M hydrochloric acid with a high SF value (SF^68^Ge/^68^Ga = 2984 ± 9.542).

The ^68^Ge source was dispersed in 0.01 M HCl, and then adsorbed onto LC-TiO_2_ adsorbent for 3 h in consideration of the previously evaluated adsorption/desorption capacity of ^68^Ge/^68^Ga. The unbounded ^68^Ge was removed from the column with 20 mL of 0.05 M HCl. After 24 h, ^68^Ga was eluted using 0.05 M HCl at a flow rate of 0.5 mL·min^−1^. The final eluted ^68^Ga-chloride was evaluated for metallic impurities, elution yield, breakthrough and fraction volume (Table 2). Based on encouraging results from the tests, we provided suggestions for future studies and the use of LC-TiO_2_ as a ^68^Ge/^68^Ga generator column material.

## 4. Materials and Methods

### 4.1. General

Low-molecular weight chitosan (MW 50,000–190,000 Da, cP 20–300) and P25 (titanium(IV) oxide, 21 nm) were obtained (Sigma-Aldrich, Saint Lousis, MO, USA). For the labeling study, DOTA and NOTA chelator were obtained from Macrocyclic^TM^. All optima grade acid solutions were purchased from Fisher Scientific (Waltham, MA, USA) and used without further purification. The ^68^Ge-chloride source was obtained from Eckert & Ziegler (Berlin, Germany). ^68^Ge/^68^Ga generator (Eckert & Ziegler, Berlin, Germany) was used for a distribution coefficient of ^68^Ga with 0.1 M HCl of an eluent. The activities of ^68^Ge and ^68^Ga were measured by a dose calibrator (Atomlab^TM^, Biodex, New York, NY, USA). For measurement of ^68^Ge breakthrough, a gamma counter (1480 WIZARD-3, Perkin Elmer, Trenton, NJ, USA) was used, and an in vivo study was conducted using a bench-top PET/X-ray system (Genesis 4, Perkin Elmer, Trenton, NJ, USA).

### 4.2. Preparation and Characterization of Chitosan-TiO_2_ Adsorbent

The most common method for synthesizing chitosan-TiO_2_ is mechanical stirring, followed by titanium metal trapped on the chitosan polymeric matrix (Figure 1). Solution I: 1.5 g of low molecule chitosan was dissolved in 5% acetic acid for 24 h. Solution II: 38.4 mL of absolute ethanol (99%) was mixed with 45 mL of TTIP dispersed in 6 mL of conc. HCl. TiO_2_ metal powder (P25) (10 g) was added to the above mixture (solution II) and allowed to homogenize using an overhead stirrer for 16 h. The premixed solution was reacted with constant stirring for 2 h. After thorough homogenization of low molecular chitosan-TiO_2_ (LC-TiO_2_), the mixture was dropped into a 28% ammonia solution using a syringe pump (needle: 27G, flow rate: 7 mL·min^−1^). The squashy LC-TiO_2_ particles were washed and filtered by gravity with distilled water and dried for 12 h at 80 °C. Finally, prebuilt LC-TiO_2_ was calcinated for 6 h at 450 °C using a muffle furnace (10 °C·min^−1^).

### 4.3. Acid Resistance Study

The acid resistance of low-molecular weight chitosan-TiO_2_ (LC-TiO_2_) was estimated by focusing on the ^68^Ga elution of extractants. The LC-TiO_2_ adsorbent was placed in a 15 mL conical tube, followed by 5 mL of hydrochloric acid at various concentrations (0.01 M to 1.0 M). Then, physical stress was applied at 200 rpm for 1 to 48 h. After batch immersing experiments, the supernatant was collected and measured for metallic impurities by ICP-MS (7500 series, Agilent Technology, Inc., Santa Clara, CA, USA).

### 4.4. Distribution Coefficients (K_d_)

The distribution coefficients of ^68^Ge and ^68^Ga with the LC-TiO_2_ were carried out with various HCl concentrations (0.01 M to 1.0 M) at 37 °C for 3 h to determine their adsorption capacity. In brief, the ^68^Ge/^68^Ga equilibrated source (Eckert & Ziegler) was used for the *K_d_* experiment.

The ^68^Ge/^68^Ga solutions (~37 kBq) were added to the LC-TiO_2_ and shook constantly. After reaching the adsorption equilibrium, adsorbents were filtered out from equilibrium solutions. The filtered solutions were determined instantly and after 24 h using a gamma counter. The calculation of the *K_d_* was carried out as
(1)$$K_{d}=\frac{(A_{i}-A_{eq})V}{mA_{eq}}\mathrm{mL}\cdot g^{-1}$$
where *K_d_* is the distribution coefficient (mL·g^−1^), *A_i_* is initial aqueous phase activity of ^68^Ge/^68^Ga before equilibration, *A_eq_* is final aqueous phase activity of ^68^Ge/^68^Ga after equilibration *V* is the volume (unit: mL) and *m* is the mass of adsorbent (grams).

### 4.5. Column Study

5 g of LC-TiO_2_ was wet-packed into a 5 mL empty column (Eichrom) with 0.05 M HCl containing 74 MBq of ^68^Ge/^68^Ga solution. ^68^Ga elution ability was tested under conditions of a flow rate of 5 mL·min^−1^ and gravity. The column was connected with a syringe pump using fluoropolymer resin tubing. After ^68^Ge/^68^Ga equilibrium, the column was rinsed with 20 mL of 0.01 M HCl to remove the unbound ^68^Ge and then was left to stand for 24 h to demonstrate its performance. The column studies were repeated to confirm reproducibility.

### 4.6. ^68^Ga Elution Profile and ^68^Ge Breakthrough

Ten fractions of 1 mL volume were eluted at 0.5 mL·min^−1^ flow rate with a syringe pump for evaluating the elution yield of Ga. After 24 h, the remaining radioactivity was measured with a gamma counter to determine the breakthrough of ^68^Ge.

### 4.7. Equilibrium Time of ^68^Ge/^68^Ga

The synthesized LC-TiO_2_ was completely soaked in 0.01 M HCl and then the supernatant was removed. ^68^Ge stock solution (in 0.01 M HCl) (3.7 MBq) was added to the activated LC-TiO_2_ adsorbent. The time to reach the ^68^Ge/^68^Ga equilibrium was evaluated by measuring the radioactivity over time for 50 h with a dose calibrator.

### 4.8. Radiolabeling and In Vivo Evaluation

Chelators (10 μM of DOTA and NOTA, respectively) were dissolved in sodium acetate (0.25 M, pH 4.5) solution. Chelator solutions were freshly prepared from stock solutions for each experiment. The ^68^Ga-chloride solution (~59 MBq·mL^−1^) was purged with inert N_2_ gas at 90 °C for 20 min (volume; <50 µL). ^68^Ga activity was added to the chelator solution (150 µL), and the reaction solutions were incubated at 100 °C and 25 °C, respectively. Samples of ^68^Ga-DOTA and NOTA measured 12 min after labeling (Stationary phase: iTLC-SG, Mobile phase: 1:1/MeOH:10% NH_4_OAc) are shown in Figure 7b,c.

For PET image scanning, balb/c mice were anesthetized with 2% isoflurane, and ^68^Ga labeled compounds were injected by intravenous tail injection. PET images were acquired 5 and 30 min after injection.

## 5. Conclusions

In summary, we demonstrated that synthesized low molecular weight chitosan-TiO_2_ can function as a column adsorbent for a ^68^Ge/^68^Ga generator. The fractions of the radionuclide generator need to be eluted and used immediately in the medical field, should not contain metallic impurities and must be used with high specific activity. Commercially available ^68^Ge/^68^Ga generators have a low labeling efficiency of ^68^Ga with chelator agents due to metallic impurities in the final ^68^Ga elution. The filter process of ^68^Ga elution is generally conducted before the manufacture of ^68^Ga-radiopharmaceuticals to avoid the effect of labeling yield by containing impure metals. We have developed a column adsorbent, which can overcome these limitations. The metallic impurities of the ^68^Ga eluate were very low and had excellent ^68^Ge/^68^Ga adsorption and desorption capacities. Based on these results, we propose the use of LC-TiO_2_ adsorbent for a ^68^Ge/^68^Ga generator system with promising elution efficiency and stability and having the capability to provide a high purity of ^68^Ga elution (>99.9%) with 10^−4^% of ^68^Ge breakthrough.

## Data Availability

Not applicable.

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
