# Peer review of "TiO_2_ Decorated Low-Molecular Chitosan a Microsized Adsorbent for a ^68^Ge/^68^Ga Generator System"

_molecules, 2021, doi:10.3390/molecules26113185_

Round 1

Reviewer 1 Report

All raised questions have been addressed and the manuscript quality has improved considerably.

Interest for scientific community and relevance would benefit from a comparision with existing Generators and not only with former work by the authors.

Author Response

Thank you for considering our article for publication in Molecules. I am grateful to you and the reviewers for the valuable suggestions provided.

Reviewer 2 Report

The article: TiO2 decorated low-molecular chitosan micro-sized adsorbent for 68Ge/68Ga generator system is an interesting and fits the subject of Molecules. Reported the column material for the 68Ge/68Ga generator with acid resistance and excellent adsorption and desorption capacity of 68Ge and 68Ga, respectively. Despite being a core element of the 68Ge/68Ga generator system, research on this has been insufficient. Therefore, we synthesized a low molecular chitosan-based TiO2 (LC-TiO2) adsorbent via a physical trapping method as a durable 68Ge/68Ga generator column material. The adsorption/desorption studies exhibit a higher separation factor of 68Ge/68Ga in the concentration range of HCl examined (0.01 M to 1.0 M). The prepared LC-TiO2 adsorbent showed acid resistance capabilities with > 93% of 68Ga elution yield and 1.6 x 10-4% of 68Ge breakthrough. In particular, the labeling efficiency of DOTA and NOTA by using the generator eluted 68Ga was quite encouraging and confirmed to be 99.65 and 99.69%, respectively. Accordingly, the resulting behavior of LC-TiO2 towards the 68Ge/68Ga adsorption/desorption capacity and stability with aqueous HCl exhibited a high potential for ion-exchange solid phase extraction for 68Ge/68Ga generator column material. The result analysis is accurate and adequate. The authors, using the appropriate equipment, have thoroughly investigated the issue. Therefore, the manuscript can be recommended for publication in Molecules.

The following points should be taken into account:

  1. It would be necessary to unify units according to the Si system
  2. Incorrect order of publication parts, e.g.: Materials and Methods
  3. What was the specific surface area and pore size of the starting TiO2 used for synthesis?
  4. Novelty elements should be better highlighted in the introduction. Papers should be cited in Introduction section; for example:

Effects of the adsorption of Ba2+ ions on the electrical double layer at the 4th group metal oxide/electrolyte interface. Adsorption Science &Technology 22 (10) (2004).

Electokinetic and adsorption properties of different titanium dioxide at the solid/solution interface. Central European Journal of Chemistry 12 (11) (2014) 1194-1205

Structure of electrical double layer at the metal oxide with proteins/NaCl electrolyte solution interface. Adsorption Science & Technology 33(6-8) (2015) 567–574

Author Response

Comments from the reviewers:
Thank you for considering our article for publication in Molecules. I am grateful to you and the reviewers for the valuable suggestions provided.
Here are responses to the reviewer comments:

Reviewer
Comments: The article: TiO2 decorated low-molecular chitosan micro-sized adsorbent for 68Ge/68Ga generator system is an interesting and fits the subject of Molecules. Reported the column material for the 68Ge/68Ga generator with acid resistance and excellent adsorption and desorption capacity of 68Ge and 68Ga, respectively. Despite being a core element of the 68Ge/68Ga generator system, research on this has been insufficient. Therefore, we synthesized a low molecular chitosan-based TiO2 (LC-TiO2) adsorbent via a physical trapping method as a durable 68Ge/68Ga generator column material. The adsorption/desorption studies exhibit a higher separation factor of 68Ge/68Ga in the concentration range of HCl examined (0.01 M to 1.0 M). The prepared LC-TiO2 adsorbent showed acid resistance capabilities with > 93% of 68Ga elution yield and 1.6 x 10-4% of 68Ge breakthrough. In particular, the labeling efficiency of DOTA and NOTA by using the generator eluted 68Ga was quite encouraging and confirmed to be 99.65 and 99.69%, respectively. Accordingly, the resulting behavior of LC-TiO2 towards the 68Ge/68Ga adsorption/desorption capacity and stability with aqueous HCl exhibited a high potential for ion-exchange solid phase extraction for 68Ge/68Ga generator column material. The result analysis is accurate and adequate. The authors, using the appropriate equipment, have thoroughly investigated the issue. Therefore, the manuscript can be recommended for publication in Molecules.
The following points should be taken into account:
1. It would be necessary to unify units according to the Si system

- This paper was written by referring to the 9th edition SI brochure.
2. Incorrect order of publication parts, e.g.: Materials and Methods

- Corrected (Manuscript line No.: 222, 234, 241, 253, 261, 265, and 270)
3. What was the specific surface area and pore size of the starting TiO2 used for synthesis?

- TiO2 (P25) is sold in the market, and it is confirmed to have a specific surface area of 35-65 m2/g. However, pore size measuring isn't easy to meet the reservation of the measuring equipment withinfive days. We ask for understanding regarding this matter.

4. Novelty elements should be better highlighted in the introduction. Papers should be cited in Introduction section;

- Inserted (Manuscript line No.: 58, Reference No.: 23, 24)